# Enhanced Anti-Tumor Effect of Folate-Targeted FA-AMA-hyd-DOX Conjugate in a Xenograft Model of Human Breast Cancer

**DOI:** 10.3390/molecules26237110

**Published:** 2021-11-24

**Authors:** Tian-tian Liao, Jiang-fan Han, Fei-yue Zhang, Ren Na, Wei-liang Ye

**Affiliations:** 1Department of Pharmaceutics, School of Pharmacy, Fourth Military Medical University, Xi’an 710032, China; TiantLiao@163.com (T.-t.L.); zfy0702@sina.cn (F.-y.Z.); 2Lab for Bone Metabolism, Key Lab for Space Biosciences and Biotechnology, School of Life Sciences, Northwestern Polytechnical University, Xi’an 710072, China; Jfanhan@mail.nwpu.edu.cn; 3Department of Epidemiology and Health Statistics, Faculty of Military Preventive Medicine, Fourth Military Medical University, Xi’an 710032, China; narentaiyang@126.com

**Keywords:** breast cancer, doxorubicin, prodrug delivery, cancer targeted therapy, biosafety

## Abstract

Folate-aminocaproic acid-doxorubicin (FA-AMA-hyd-DOX) was firstly synthesized by our group. It was indicated that FA-AMA-hyd-DOX was pH-responsive, and had strong cytotoxicity on a folate receptor overexpressing cell line (KB cells) in vitro. The aim of our study was to further explore the potential use of FA-AMA-hyd-DOX as a new therapeutic drug for breast cancer. The cellular uptake and the antiproliferative activity of the FA-AMA-hyd-DOX in MDA-MB-231 cells were measured. Compared with DOX, FA-AMA-hyd-DOX exhibited higher targeting ability and cytotoxicity to FR-positive tumor cells. Subsequently, the tissue distribution of FA-AMA-hyd-DOX was studied, and the result confirmed that DOX modified by FA can effectively increase the selectivity of drugs in vivo. After determining the maximum tolerated dose (MTD) of FA-AMA-hyd-DOX in MDA-MB-231 tumor-bearing nude mice, the antitumor effects and the in vivo safety of FA-AMA-hyd-DOX were systematically evaluated. The data showed that FA-AMA-hyd-DOX could effectively increase the dose of DOX tolerated by tumor-bearing nude mice and significantly inhibit MDA-MB-231 tumor growth in vivo. Furthermore, FA-AMA-hyd-DOX treatment resulted in almost no obvious damage to the mice. All the positive data suggest that FA-targeted FA-AMA-hyd-DOX is a promising tumor-targeted compound for breast cancer therapy.

## 1. Introduction

Cancer is one of the major health problems of the 21st century, and breast cancer is the most common malignant tumor in women [1]. At present, the implementation of the new therapy has greatly improved the survival rate of patients with breast cancer. However, breast cancer remains one of the leading causes of death among women [2]. Therefore, it is urgent to explore effective treatments to improve the quality of life of breast cancer patients.

In order to improve the anti-tumor effects of drugs, many drug delivery systems have been developed successfully. However, in most cases, the compositions and preparations of these nanodelivery systems are extremely complex and challenging. Difficulties, including in the synthesis and purification of materials, hamper large-scale production [3]. In addition, most of the carriers and their degradation products that are currently used have no therapeutic effect, and some may even cause obvious side effects, including cardiovascular effects, inflammation and so on [4].

The use of prodrugs to modify lead compounds can improve the bioavailability, stability, toxicity and long-term efficacy of drugs [5,6]. In cancer treatment research, scientists have designed and synthesized a number of anti-tumor prodrugs which can greatly improve the current clinical efficacy of chemotherapeutic drugs that are toxic, non-selective and poor in physical properties [7,8]. Zhang [9] synthesized a polyethylene glycol-loaded, biodegradable curcumin (Cur) prodrug to address the problem of curcumin’s water solubility and enhance its efficacy. The results showed that the Cur prodrug has a simple synthesis method, a high synthesis rate and a slow-release anti-tumor effect, making it a good prospect for further research and development. In addition, prodrugs which can overcome drug use obstacles, enhance chemical and metabolic stability, increase oral or local drug absorption, enhance blood–brain barrier permeability, prolong action time, improve bioavailability and reduce adverse reactions, have become favored and are widely accepted.

However, for most prodrugs, their release in tumor cells is not satisfactory, which to some extent hinders their specific anti-tumor efficacy. To conquer these challenges, environment responsive delivery systems have been designed to improve drug release in tumor cells. Of the related stimuli, pH is the most frequently used, as in different tissues and organelle the pH values vary dramatically [10,11]. Hydrazone bonds are known to be stable at pH 7.4 and degrade at low pH values (<5.0), and they have been successfully used to generate a variety of pH-sensitive conjugates for specific drug delivery [12]. In our previous study, we synthesized a pH-responsive FA-targeted prodrug FA-AMA-hyd-DOX. Owing to the presence of the FA moiety, FA-AMA-DOX can largely be internalized by the folate receptor overexpressing tumor cells via FR-mediated endocytosis. As the hydrazone linkage is acid labile, the release of DOX is increased in the acidic environment of the tumor cell lysosomes. In addition, the prodrug can exert a cytotoxic effect effectively and has good application prospects [13].

In this experiment, in order to further study the potential use of FA-AMA-hyd-DOX as a new cancer therapeutic, we compared the uptake and the antiproliferative activity of FA-AMA-hyd-DOX and DOX in MDA-MB-231 cells, which were known to significantly overexpress the folate receptor [14,15]. Subsequently, the distributions of FA-AMA-hyd-DOX and DOX in tumor tissues and normal organs were compared by intravenous injection in tumor-bearing mice. After determining the maximum tolerated dose (MTD) of FA-AMA-hyd-DOX in MDA-MB-231 tumor-bearing nude mice, we compared the antitumor effects of FA-AMA-hyd-DOX and DOX. At the same time, the in vivo safety of FA-AMA-hyd-DOX was systematically evaluated.

## 2. Results and Discussion

Breast cancer is one of the tumor types where DOX has shown high anticancer activity [16]. However, the application of DOX in the clinic has been limited by its serious cardiotoxicity. In our previous study, we successfully synthesized FA-AMA-hyd-DOX. Its structure is shown in Figure 1A. It was proven that FA-AMA-hyd-DOX kept stable at a neutral pH, but DOX could be released when the FA-AMA-hyd-DOX was incubated in a low pH value (pH = 5.0) medium. This release profile implied that FA-AMA-hyd-DOX is stable in the blood during in vivo circulation, whereas DOX would be released quickly after FA-AMA-hyd-DOX is selectively internalized by the acid lysosomes. Owing to the presence of the FA moiety, FA-AMA-hyd-DOX showed high cytotoxicity in the FR over-expressed tumor cells [13]. In order to further study the antitumor activity of FA-AMA-hyd-DOX, its cellular uptake and cytotoxicity in MDA-MB-231 cancer cells were studied, and its therapeutic efficacy in a female athymic nude mouse tumor model bearing MDA-MB-231 cancer cells was also evaluated.

### 2.1. Cellular Uptake of FA-AMA-hyd-DOX

Drug delivery efficiency is closely related to cellular uptake. High cellular uptake of a drug can result in effective treatment efficacy [17,18]. Previous studies have shown that FA-AMA-hyd-DOX can be efficiently ingested by KB cells with high folate receptor expression, which can promote tumor-targeted drug delivery [13]. Herein, the cellular uptake of DOX and its derivatives was observed in MDA-MB-231 cancer cells by the LSCM. As shown in Figure 1B,D, the intensity of fluorescence in DOX and its derivative groups increased as the incubation time prolonged to 2 h. Not surprisingly, the exposure of MDA-MB-231 cancer cells to FA-AMA-hyd-DOX resulted in greater cellular uptake compared with DOX and AMA-hyd-DOX, presumably because of FA-mediated specific endocytosis. Interestingly, after the exposure of FA-AMA-hyd-DOX to MDA-MB-231 cancer cells for 2 h, the DOX red fluorescence almost localized completely in the nucleus, where DOX exhibited antitumor activity. This result indicates that FA-AMA-hyd-DOX was taken up efficiently by MDA-MB-231 cells, and then DOX could be rapidly released from acid-sensitive FA-AMA-hyd-DOX, diffusing into the nucleus to exert its antitumor activity [19]. Hence, FA-AMA-hyd-DOX not only significantly enhanced DOX uptake by cancer cells through the folate receptor, but also effectively released DOX in the cytoplasm, both of which are critical to DOX’s cytotoxicity.

The cellular uptake of DOX and its derivatives was further evaluated by flow cytometry. As shown in Figure 1C,E, the intensity augmented as incubation time increased. The cellular uptake of FA-AMA-hyd-DOX was greater than the uptake of DOX and that of AMA-hyd-DOX. These data are consistent with the results of the confocal experiment, and we can draw the conclusion that the FA-decorated FA-AMA-hyd-DOX has the ability to convey DOX to the targeted cancer cells, which also indicates that FA-AMA-hyd-DOX should have a stronger anti-tumor effect.

Figure 2 shows the distribution of DOX red fluorescence in the cytoplasm and nucleus after MDA-MB-231 cancer cells were incubated with FA-AMA-hyd-DOX. There was plenty of DOX red fluorescence localized in cytoplasm after MDA-MB-231 cells were incubated for 30 min, but little DOX red fluorescence localized in nucleus. However, a large amount of DOX red fluorescence was localized in nucleus after 2 h incubated, and the DOX red fluorescence in cytoplasm was significantly reduced. These results clearly indicate that the folate receptor-mediated endocytosis process promotes the cellular uptake of FA-AMA-hyd-DOX. Furthermore, when FA-AMA-hyd-DOX was incubated with MDA-MB-231 cells, it was mainly localized in endolysosomes at 30 min. FA-AMA-hyd-DOX showed a distinct nucleus distribution of DOX at 2 h. This is because FA-AMA-hyd-DOX can be dissociated in the endolysosomes, subsequently speeding up the release of DOX, which then enters into the nucleus with prejudice to exert cytotoxic effects.

### 2.2. Cytotoxicity of FA-AMA-hyd-DOX

The cytotoxicity of DOX and its derivatives against MDA-MB-231 cancer cells and HUVEC normal cells was evaluated via MTT assay. As shown in Figure 3A–D, for the tumor cell lines, all drugs showed dose responsive anti-tumor activity at different concentrations. In the tumor cells, FA-AMA-hyd-DOX exhibited no significantly improved proliferation inhibition effects in comparison to DOX at an equivalent concentration after incubation for 12 h. On the other hand, the same concentration of DOX exhibited a lower tumor-inhibition rate than the FA-AMA-hyd-DOX after incubation for 24 h. This phenomenon was mainly due to the fact that DOX needed a certain amount of time to dissociate from the prodrug FA-AMA-hyd-DOX. Interestingly, the FA-AMA-hyd-DOX showed a relatively low cell inhibition ratio to HUVEC normal cells compared with that of the DOX at the same concentration. These results demonstrate that FA-AMA-hyd-DOX showed low cytotoxicity to normal cells and selective therapeutic efficacy to tumor cells.

### 2.3. Apoptosis of FA-AMA-hyd-DOX

As shown in Figure 3E,F, all drugs induced apoptosis in MDA-MB-231 cancer cells in a time-dependent manner. When MDA-MB-231 cancer cells were incubated with the drugs for 12 h, there was no difference in the apoptotic cell ratio among the DOX, AMA-hyd-DOX and FA-AMA-hyd-DOX. FA-AMA-hyd-DOX induced significantly more apoptosis compared with DOX and AMA-hyd-DOX after 24 h of incubation.

Activation of caspase-3 is a critical step in the execution of apoptosis [20]. Thus, the caspase-3 levels in MDA-MB-231 cancer cells were measured after treatments with DOX and its derivatives. As shown in Figure 3G,H, compared with the control, DOX and its derivatives significantly increased the caspase-3 level in MDA-MB-231 cancer cells in a time-dependent manner. Interestingly enough, FA-AMA-hyd-DOX exhibited no significant effect on the caspase-3 level as compared with DOX and AMA-hyd-DOX after 12 h of incubation. FA-AMA-hyd-DOX increased the caspase-3 level much more compared with DOX and AMA-hyd-DOX after 24 h of incubation. The above data are consistent with the cytotoxicity of DOX, AMA-hyd-DOX and FA-AMA-hyd-DOX on MDA-MB-231 cancer cells.

### 2.4. MTD of FA-AMA-hyd-DOX

The in vivo biocompatibility of the drug was evaluated through dose studies to determine the maximum tolerated dose (MTD) of the drug itself [21,22]. In this study, there was no remarkable reduction in body weight or any other toxic reaction in mice treated with FA-AMA-hyd-DOX at doses of 0.92–18.4 µmol/kg within 2 weeks after its injection. The 36.8 µmol/kg FA-AMA-hyd-DOX treatment caused body weight loss (5/5), and the mice needed to be removed from the study. The 9.2 µmol/kg DOX treatment caused body weight reduction (3/5) but no deaths. The 18.4–36.8 µmol/kg DOX treatment resulted in a quick reduction in the body weight (5/5), and all animals died within 2 weeks. The AMA-hyd-DOX showed a similar result as the DOX. Therefore, the MTD of DOX, AMA-hyd-DOX and FA-AMA-hyd-DOX in female athymic nude mice were, for a single injection, approximately 9.2, 9.2 and 18.4 µmol/kg, respectively.

### 2.5. Biodistribution Study of FA-AMA-hyd-DOX

The tissue distribution was studied in female athymic nude mice bearing MDA-MB-231 tumors and analyzed by Caliper IVIS Lumina II in vivo imaging (Caliper Life Science, Hopkinton, MA, USA). The fluorescence intensity was quantitatively analyzed by Living Image 4.2 soft-ware (dark yellow represents high fluorescence intensity; dark red means relatively weak fluorescence intensity). As shown in Figure 4A,B, DOX and AMA-hyd-DOX were extensively accumulated in all the tissues, especially the heart, liver and kidney. However, compared with DOX and AMA-hyd-DOX, FA-AMA-hyd-DOX significantly increased the DOX concentration in tumors, and greatly decreased the DOX concentrations in the other tissues. According to the research, the biggest side effect of DOX in clinical applications is that it can produce serious cardiotoxicity [23,24]. When FA-AMA-hyd-DOX was given to tumor-bearing nude mice, the presence of DOX was hardly observed in heart tissue. The results suggested that the application of FA-AMA-hyd-DOX may weaken or eliminate the cardiotoxicity of DOX. Besides, no significant difference was observed between AMA-hyd-DOX and DOX accumulation in any tissue. Those results further confirmed that DOX modified by FA can effectively increase the selectivity of drugs in vivo.

### 2.6. In Vivo Antitumor Activity of FA-AMA-hyd-DOX

The results are shown in Figure 5. FA-AMA-hyd-DOX has the best antitumor effect, as shown by the smallest tumor volume (Figure 5A,B). DOX showed the weakest antitumor effect during the therapy, which should be attributed to the quick excretion by glomerular filtration [25]. The survival times of tumor-bearing mice were evaluated, and the results are shown in Figure 5C and Table 1. The median survival time of mice treated with FA-AMA-hyd-DOX (38 days) was remarkably longer than that of mice treated with DOX (26 days, *p* = 0.0255). From the above results, we can draw the conclusion that the targeted FA-AMA-hyd-DOX had an obvious antitumor effect, making the tumor-bearing nude mice have a longer survival time.

Histological analysis was carried out to further evaluate the antitumor effect. Tumor specimens were collected for hematoxylin and eosin (H&E) staining and TUNEL analysis (Figure 5D,E). Compared with the saline groups, the drug-treated groups showed smaller cells and wider intracellular spaces; and even lysis of nuclei emerged, especially in the FA-AMA-hyd-DOX group. This indicates that there were some inflammatory and cytotoxic responses in the tumor treated with FA-AMA-hyd-DOX. Then the TUNEL images were employed to observe the apoptosis. As shown in Figure 5E, the DNA fragmentations were stained by a green fluorescent probe. The fluorescence intensity of the FA-AMA-hyd-DOX-treated tumor was the strongest among all of the experiment groups, which was consistent with its superiority in antitumor efficiency in in vivo and histopathological results.

### 2.7. Security Evaluation of FA-AMA-hyd-DOX

The in vivo security of antitumor drugs is a critical evaluation index for clinical chemotherapies, which is directly linked to the survival of malignancy patients. In this study, the safety of drugs was systematically assessed through the detection of body weight change, and analyses of hematological parameters. As shown in Figure 5F, significant body weight loss was observed in the DOX and AMA-hyd-DOX groups due to the serious toxicity and side effects, whereas just a slight decrease in body weight was observed in the FA-AMA-hyd-DOX group. The results revealed that the FA-targeted prodrug exhibited improved safety in vivo.

It is reported that the application of DOX can easily lead to acute cardiotoxicity and nephrotoxicity [26,27]. As shown in Figure 5G,H, DOX caused significant cardiotoxicity, with CK and LDH values obviously larger than those of the control group. In contrast, the CK and LDH values caused by FA-AMA-hyd-DOX injection were far lower than those of DOX and AMA-hyd-DOX, indicating lower cardiotoxicity. Similarly, the BUN and Cr values of FA-AMA-hyd-DOX were similar to those of the control group, indicating negligible nephrotoxicity. In contrast, DOX caused severe kidney failure, as evidenced by the larger BUN and Cr values. All of the above data demonstrate that FA-AMA-hyd-DOX was safe for the mice and has clinical application potential.

## 3. Materials and Methods

### 3.1. Chemical Products

Folate-aminocaproic acid-doxorubicin (FA-AMA-hdy-DOX) was produced by our lab as previously described [13]. Folic acid (FA) was purchased from Sigma-Aldrich (St. Louis, MO, USA). DOX hydrochloride was purchased from Hisun Pharmaceutical Co. (Hangzhou, China).

### 3.2. Cell Lines

The human breast cancer cells, MDA-MB-231 (over express FR) and human umbilical vein vessel endothelial cells, HUVECs, were obtained from the Institute of Biochemistry and Cell Biology, Chinese Academy of Science, Shanghai.

### 3.3. Intracellular DOX Analyses

The cellular uptakes of DOX and its derivatives by MDA-MB-231 cancer cells were detected by both laser scanning confocal microscopy (LSCM, Olympus FV10-ASW, Tokyo, Japan) and flow cytometry (FCM, Becton Dickinson FACScan, Franklin Lakes, NJ, USA).

#### 3.3.1. LSCM

Briefly, the cells were seeded in 24-well plates at a density of 1.2 × 10^4^ cells/mL, and incubated in the medium of folate-free RPMI-1640. After 24 h of incubation, the medium was replaced by fresh medium. Then, DOX or one of its derivatives was added to each well at equivalent doxorubicin concentration of 10 µg/mL. After incubation for 30 min or 2 h, the images of cells were obtained by using a LSCM (Olympus FV10-ASW, Tokyo, Japan). The intracellular fluorescence intensity was calculated by using ImageJ software (Media Cybernetics, MD, USA).

#### 3.3.2. FCM

The cells were seeded in 6-well plates at a density of 1.5 × 10^6^ cells/mL, and incubated in the medium of folate-free RPMI-1640. After 24 h of incubation, the medium was replaced by fresh medium. Then, DOX or a derivative was added to each well at an equivalent doxorubicin concentration of 10µg/mL. The cells were continued cultured for 30 min or 2 h. Cells without treatment were used as control. Finally, the cells were harvested for flow cytometry (FCM, Becton Dickinson FACScan, Franklin Lakes, NJ, USA) analysis.

### 3.4. Cytotoxicity Assays

Briefly, the cells were seeded in 96-well plates at a density of 1.5 × 10^4^ cells/mL, and incubated in the medium of folate-free RPMI-1640. After 24 h of incubation, the medium was replaced by fresh medium. Then different concentrations of DOX and its derivatives were added for 12 h or 24 h. The percentage of cell viability was determined at 570 nm by using a microplate reader (Bio-Rad Laboratories, Inc., Richmond, CA, USA).

### 3.5. Apoptosis Analyses

The cells were seeded into 6-well plates at a density of 2 × 10^5^ cells/mL, and incubated in the medium of folate-free RPMI-1640 for 24 h. Then, the culture medium was replaced by fresh medium containing DOX and its derivatives (the equivalent DOX concentration was 20 μg/mL). After being cultured for 12 h or 24 h, they were analyzed by a Becton Dickinson FACScan (excitation at 488 nm) equipped with Cell Quest software (BD Biosciences, Franklin Lakes, Franklin Lakes, NJ, USA).

In addition, the effects of DOX and its derivatives on caspase-3 level in MDA-MB-231 cancer cells were measured by using the caspase-3 detection kit (Beyotime Institute of Biotechnology, Nanjing, China). The MDA-MB-231 cancer cells were cultured in 96-well plates at a density of 2 × 10^4^ cells/mL and incubated for 24 h. Then, the culture medium was replaced by fresh medium containing DOX and its derivatives (the equivalent DOX concentration was 20 μg/mL). After being cultured for 12 h or 24 h, the caspase-3 level was determined at 405 nm on a microplate reader according to the manufacturer’s instructions.

### 3.6. Animal Experiments

All animal procedures were performed in according with protocols approved by the Animal Care and Use Committee of Fourth Military Medical University. The female athymic nude mice were obtained from Experimental Animal Center of Fourth Military Medical University.

#### 3.6.1. Maximum Tolerated Dose (MTD) Studies

DOX or a derivative (the equivalent dose of the DOX) was dissolved in normal saline and then intravenously administered to the normal female athymic nude mice at doses of 0.92, 4.6, 9.2, 18.4, and 36.6 µmol/kg (*n* = 5). The survival rate and body weight were recorded every other day for 2 weeks. The definition of MTD is the dose that reduces the weight of mice by no more than 15% and no death during 2 weeks [28].

#### 3.6.2. Biodistribution Analyses

Tumor-bearing nude mice were intravenously injected with DOX or a derivative (the equivalent dose of the DOX was 2.5 mg/kg). The nude mice were euthanized 12 h later. Finally, the tumor tissue, heart, liver, spleen, lung and kidney were collected to observe the fluorescence intensity by the Caliper IVIS Lumina II in vivo image (Caliper Life Science, Hopkinton, MA, USA). The fluorescence intensity was quantitatively analyzed by Living Image 4.2 software (PerkinElmer, Health Sciences, Waltham, MA, USA).

#### 3.6.3. Anti-Tumor Activity Analyses

Tumor-bearing nude mice were randomly divided into four groups (5 mice per group). Then the mice were intravenously injected with normal saline, DOX, AMA-hyd-DOX or FA-AMA-hyd-DOX (the equivalent dose of the DOX was 2.5 mg/kg). The mice were treated every seven days for a total of three times.

#### 3.6.4. Histopathological Analyses

The tumor tissues were excised and stained with haematoxylin and eosin to observe histopathological changes.

#### 3.6.5. In Situ Cell Apoptosis Assays

The tumor tissues were excised to detect apoptosis by TUNEL.

#### 3.6.6. Organ Damage Assays

Tissue damage was detected with enzyme-linked immunosorbent assay (ELISA) kits (Shanghai Lichen Biotechnology Co., Ltd., Shanghai, China). The blood of each mouse was collected and then the biochemical indexes including creatine kinase (CK), lactate dhydrogenase (LDH), blood urea nitrogen (BUN) and creatinine (Cr) were detected following the manufacturer’s instruction.

### 3.7. Statistical Analysis

All data were processed and analyzed by GraphPad Prism 8 software (GraphPad Prism, San Diego, CA, USA). The statistical significances were evaluated by *t*-test of the software and *p* < 0.05 was considered significant.

## 4. Conclusions

Compared with DOX, FA-AMA-hyd-DOX exhibited higher targeting ability and less cytotoxicity to FR-positive tumor cells. In addition, FA-AMA-hyd-DOX could significantly inhibit MDA-MB-231 tumor growth in vivo due to the enhanced accumulation at the tumor site, and increased intracellular DOX release. Furthermore, FA-AMA-hyd-DOX exhibited almost no damage to the mice. All the positive data suggest that the FA-targeted FA-AMA-hyd-DOX is a promising tumor-targeted compound for cancer therapy.

## Figures and Tables

**Figure 1 molecules-26-07110-f001:**
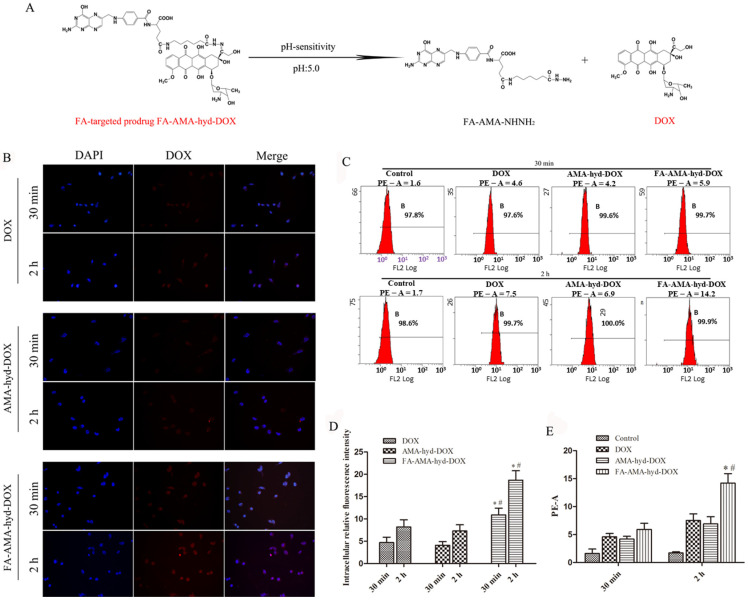
The cellular uptakes of DOX and its derivatives. (**A**) Schematic illustration of the FA-targeted and pH-responsive FA-AMA-hyd-DOX with enhanced tumor cell selectivity and controlled drug release. (**B**) Confocal images of MDA-MB-231 cancer cells incubated with DOX, AMA-hyd-DOX and FA-AMA-hyd-DOX for 30 min or 2 h at 37 °C. (**C**) Flow cytometry images of MDA-MB-231 cancer cells incubated with DOX, AMA-hyd-DOX and FA-AMA-hyd-DOX for 30 min or 2 h at 37 °C. (**D**) The quantitative analysis of DOX fluorescence intensity detected by LSCM. ** p* < 0.05 vs. DOX at the same time point, *# p* < 0.05 vs. AMA-hyd-DOX at the same time point. (**E**) The quantitative analysis of the MDA-MB-231 cancer cells detected by flow cytometry. * *p* < 0.05 vs. DOX at the same time point, # *p* < 0.05 vs. FA-AMA-hyd-DOX at 30 min point.

**Figure 2 molecules-26-07110-f002:**
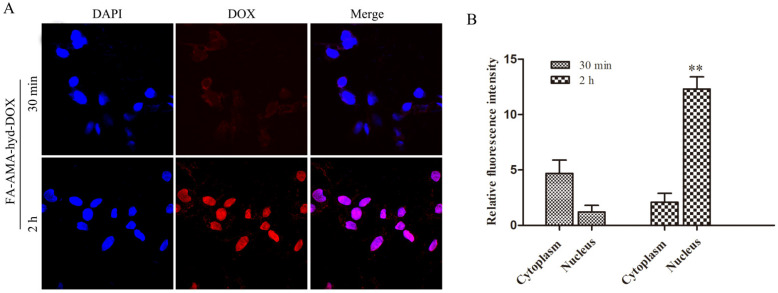
The distribution of DOX in cytoplasm and nucleus after MDA-MB-231 cancer cells were incubated with FA-AMA-hyd-DOX for 30 min or 2 h. (**A**) The semi-quantitative analysis of DOX relative fluorescence intensity in cytoplasm and nucleus. (**B**) The equivalent DOX concentration in cytoplasm and nucleus. ** *p* < 0.01 vs. nucleus with the same treatment, *n* = 3.

**Figure 3 molecules-26-07110-f003:**
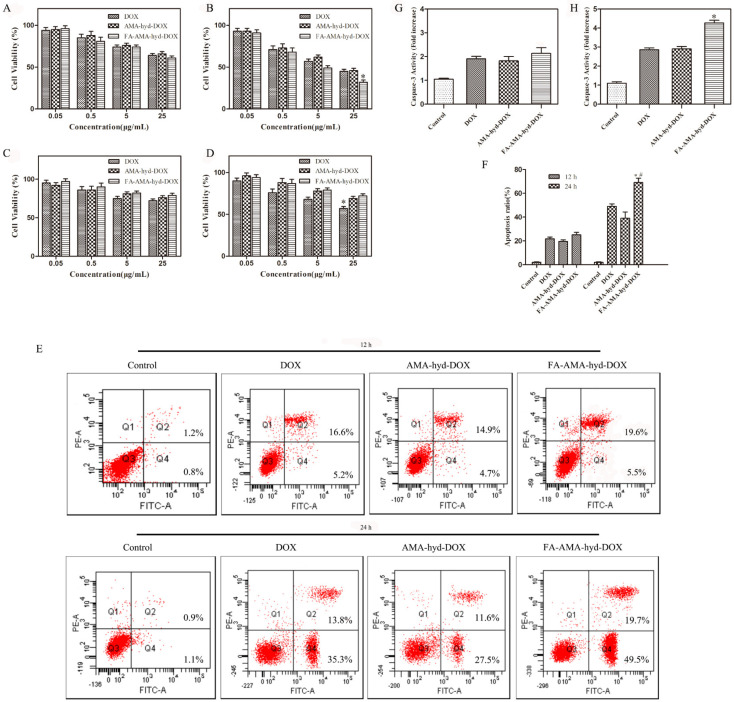
The cytotoxicity of DOX and its derivatives. MDA-MB-231 cancer cells (**A**,**B**) or HUVEC normal cells (**C**,**D**) after 12 h (**A**,**C**) or 24 h (**B**,**D**) incubations detected by MTT. * *p* < 0.05 vs. DOX at the same concentration of DOX. (**E**) Apoptosis of MDA-MB-231 cancer cells treated with DOX and its derivatives for 12 and 24 h at 37 °C. (**F**) The apoptosis ratio of MDA-MB-231 cancer cells. * *p* < 0.05 vs. DOX at the same time point, # *p* < 0.05 vs. FA-AMA-hyd-DOX at 30 min point. (**G**,**H**) The caspase-3 activity of the MDA-MB-231 cancer cells incubated with DOX and its derivatives for 12 and 24 h at 37 °C. * *p* < 0.05 vs. DOX at the same time point.

**Figure 4 molecules-26-07110-f004:**
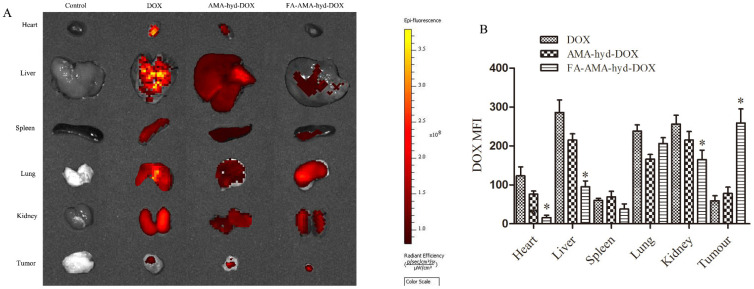
Biodistribution study of DOX and its derivatives. (**A**) The DOX biodistribution in tumor-bearing nude mice after 12 h intravenous injection of DOX, AMA-hyd-DOX and FA-AMA-hyd-DOX. (**B**) The semiquantitative analysis of ex vivo images for DOX distribution in isolated organs detected by living image system. * *p* < 0.05 vs. DOX in the same organ tissue.

**Figure 5 molecules-26-07110-f005:**
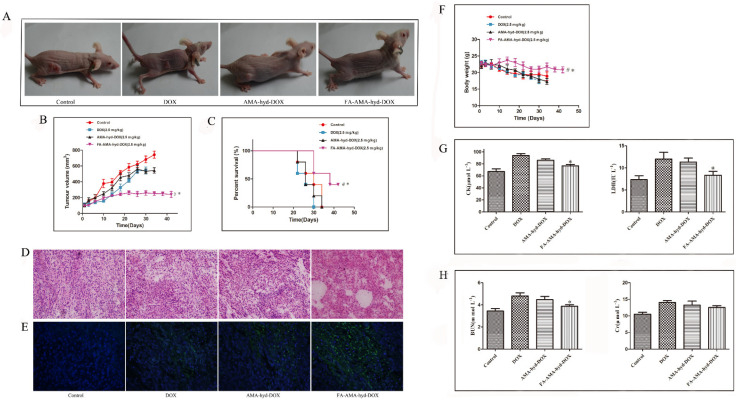
The in vivo anti-tumor activity and security evaluations of DOX and its derivatives in tumor-bearing nude mice. (**A**) Tumor-bearing nude mice recorded by camera at the end of treatment. (**B**) Tumor inhibition rate over each treatment regimen. (**C**) Survival curve over each treatment regimen. * *p* < 0.05 vs. DOX; # *p* < 0.05 vs. control, *n* = 5. (**D**) HE analysis of tumor tissues using confocal microscopy at the end of the experiments. Magnification: 200×. (**E**) TUNEL analysis of tumor tissues using confocal microscopy at the end of the experiments. Magnification: 200×. (**F**) Body weight changes over the treatment regimen. (**G**) Evaluations of heart-related CK and LDH. * *p* < 0.05 vs. DOX; *# p* < 0.05 vs. control, *n* = 5. (**H**) Evaluations of kidney-associated BUN and Cr. * *p* < 0.05 vs. DOX; *# p* < 0.05 vs. control, *n* = 5.

**Table 1 molecules-26-07110-t001:** Statistical analysis of the survival of tumor-bearing nude mice.

Treatment Group	Median Survival Time (d)	Maximal Survival Time (d)	*p*
Control	30	34	-
DOX	26	30	0.1358
AMA-hyd-DOX	26	34	0.2948
FA-AMA-hyd-DOX	38	42	0.0255 *

* Compared with DOX.

## Data Availability

Not applicable.

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
