# Peer review of "Enhanced Anti-Tumor Effect of Folate-Targeted FA-AMA-hyd-DOX Conjugate in a Xenograft Model of Human Breast Cancer"

_molecules, 2021, doi:10.3390/molecules26237110_

Round 1
Reviewer 1 Report
It is worth reading the article "Enhanced Anti-Tumor Effect of Folate-Targeted FA -AMA-hyd-DOX Conjugate in a Xenograft Model of Human Breast Cancer" by Tian-tian Liao1, Jiang-fan Han2, Fei-yue Zhang1, Ren Na3 , Wei-liang Ye1, * but it is difficult to understand the article because the authors assume that the reader has detail knowledge. It should be clearly stated at the beginning what it is about, namely that the drug doxorubicin enters the cell via folic acid receptors which particular cancer cell express to an increased extent. It is also important to emphasize that the pH-sensitive bond in the substance FA-AMA-hyd-Dox releases DOX after absorption into the (acidic) lysosomes because the pH of a tumor cell is (or can be) alkaline (Stubbs M, Veech RL. Griffiths JR, Adv Enzyme Regul. 1995; 35: 101-15). A further criticism is that the images and figures are not suitable for a thorough study of the article, they are simply too small. I ask the authors to revise their valuable article because of the points of criticism mentioned. In conclusion: Perhaps a formula scheme would be valuable that shows transport through the membrane, release of DOX in the lysosomes and enrichment of DOX in the cytoplasm or nucleus.
A few more specific comments:
is it correct that (Table 1) the animals treated with DOX and AMA-hyd-DOX die sooner than the untreated control animals?
Line 66: .... "the prodrug is inactive in blood circulation and normal tissues, but is active in tumor cells".
Since the pH value in tumor cells is or can be alkaline, but the environment of the tumor cell sacidic houldn't the word "cell" be replaced by "tissue"?
Line 101: .... presumably because of FA-mediated specific endocytosis.
Why endocytosis? and not FA mediated carrier mediated transport? Is FA-AMA-hyd-DOX enclosed in particles (nanoparticles)?
Line 177: ... "Therefore, the MTD of DOX, AMA-hyd-DOX and FA-AMA-hyd-DOX in female athymic nude mice was, for a single injection, approximately 5, 5 and 10 mg / kg , respectively ".
For comparability, please also specify doses in µmol/kg
Author Response
Cover letter
Dear editors and reviewers,
Thank you very much for your letter and advice. We have revised the paper (Enhanced anti-tumor effect of folate-targeted FA -AMA-hyd-DOX conjugate in a xenograft model of human breast cancer, Manuscript ID: molecules-1462525), and would like to re-submit it for your consideration. We have addressed the comments raised by the reviewers, and the amendments are highlighted in red in the revised manuscript. Because of your suggestions, the revised article becomes better and readers can get more valuable information. We hope that the revision is acceptable, and I look forward to hearing from you soon.
With best wishes,
Yours sincerely,
Weiliang Ye
E-mail: yaojixue@fmmu.edu.cn
Comments and Suggestions for Authors,
It is worth reading the article "Enhanced Anti-Tumor Effect of Folate-Targeted FA -AMA-hyd-DOX Conjugate in a Xenograft Model of Human Breast Cancer" by Tian-tian Liao1, Jiang-fan Han2, Fei-yue Zhang1, Ren Na3 , Wei-liang Ye1, * but it is difficult to understand the article because the authors assume that the reader has detail knowledge. It should be clearly stated at the beginning what it is about, namely that the drug doxorubicin enters the cell via folic acid receptors which particular cancer cell express to an increased extent. It is also important to emphasize that the pH-sensitive bond in the substance FA-AMA-hyd-Dox releases DOX after absorption into the (acidic) lysosomes because the pH of a tumor cell is (or can be) alkaline (Stubbs M, Veech RL. Griffiths JR, Adv Enzyme Regul. 1995; 35: 101-15). A further criticism is that the images and figures are not suitable for a thorough study of the article, they are simply too small. I ask the authors to revise their valuable article because of the points of criticism mentioned. In conclusion: Perhaps a formula scheme would be valuable that shows transport through the membrane, release of DOX in the lysosomes and enrichment of DOX in the cytoplasm or nucleus.
Point 1:〝It should be clearly stated at the beginning what it is about, namely that the drug doxorubicin enters the cell via folic acid receptors which particular cancer cell express to an increased extent. It is also important to emphasize that the pH-sensitive bond in the substance FA-AMA-hyd-Dox releases DOX after absorption into the (acidic) lysosomes because the pH of a tumor cell is (or can be) alkaline〞
Response 1: Thank you for your advice. The detailed description has been added in the manuscript.
〝In our previous study, we synthesized a pH-responsive FA-targeted prodrug FA-AMA-hyd-DOX. Owing to the presence of the FA moiety, FA–AMA–hyd-DOX can largely be internalized by the folate receptor overexpressing tumor cells via FR-mediated endocytosis. Because the hydrazone linkage is acid labile, the release of DOX increased in the acidic environment of the tumor cell lysosomes.〞
Point 2:〝A further criticism is that the images and figures are not suitable for a thorough study of the article, they are simply too small〞
Response 2: Thank you for your advice. All images in the article have been reprocessed and uploaded.
Point 3:〝Perhaps a formula scheme would be valuable that shows transport through the membrane, release of DOX in the lysosomes and enrichment of DOX in the cytoplasm or nucleus〞
Response 3: Thank you for your advice. We analyzed the fluorescence intensity of FA–AMA–hyd-DOX entering the cytoplasm and nucleus at different time periods to observe the process of FA–AMA–hyd-DOX being transported through the cell membrane, lysosomes releasing DOX, and enriching DOX in the cytoplasm or nucleus. As shown below. This part research content has been added to the article.
Fig.1 showed the distribution of DOX red fluorescence in cytoplasm and nucleus after MDA-MB-231 cancer cells were incubated with FA-AMA-hyd-DOX. There was plenty of DOX red fluorescence localized in cytoplasm after MDA-MB-231 cells were incubated for 30 min, while little DOX red fluorescence localized in nucleus. However, a large amount of DOX red fluorescence was localized in nucleus after 2 h incubated, and the DOX red fluorescence in cytoplasm was significantly reduced. These results clearly indicate that the folate receptor-mediated endocytosis process promotes the cellular uptake of FA-AMA-hyd-DOX. Furthermore, when FA-AMA-hyd-DOX was incubated with MDA-MB-231 cells, it was mainly localized in endolysosomes at 30 min. FA-AMA-hyd-DOX showed a distinct nucleus distribution of DOX at 2 h. This is because FA-AMA-hyd-DOX can be dissociated in the endolysosomes, subsequently faster release of DOX and then massively entered into the nucleus to exert cytotoxic effects.
A few more specific comments:
Point 4: is it correct that (Table 1) the animals treated with DOX and AMA-hyd-DOX die sooner than the untreated control animals?
Response 4: Yes, this is the result of experimental analysis. In vivo security of antitumor drugs is a critical evaluation index for clinical chemotherapy, which is directly linked to the survival of malignancy patients. It is reported that the application of DOX can easily lead to acute cardiotoxicity and nephrotoxicity [1,2]. In the results of our experiment, both DOX and AMA-hyd-DOX can cause dramatically weight loss in nude mice and cause serious cardiotoxicity and nephrotoxicity. This is probably why the animals treated with DOX and AMA-hyd-DOX dies sooner than the untreated control animals. The two research groups, working independently, reached similar conclusions [3,4].
- Wang, Y.; Chao, X.; Ahmad, F.; Shi, H.; Mehboob, H.; Hassan, W. Phoenix dactylifera Protects against Doxorubicin-Induced Cardiotoxicity and Nephrotoxicity. Cardiol Res Pract 2019, 2019, 7395239.
- Zhu, L.; Lin, M. The synthesis of Nano-Doxorubicin and its anticancer effect. Anticancer Agents Med Chem 2020.
- Alibolandi, M.; Abnous, K.; Sadeghi, F.; Hosseinkhani, H.; Ramezani, M.; Hadizadeh, F. Folate receptor-targeted multimodal polymersomes for delivery of quantum dots and doxorubicin to breast adenocarcinoma: In vitro and in vivo evaluation. Int J Pharm 2016, 500, 162-178.
- Shi, C.; Guo, X.; Qu, Q.; Tang, Z.; Wang, Y.; Zhou, S. Actively targeted delivery of anticancer drug to tumor cells by redox-responsive star-shaped micelles. Biomaterials 2014, 35, 8711-8722.
Point 5: Line 66: .... "the prodrug is inactive in blood circulation and normal tissues, but is active in tumor cells". Since the pH value in tumor cells is or can be alkaline, but the environment of the tumor cells acidic houldn't the word "cell" be replaced by "tissue"?
Response 5: Thank you for your advice. This description is not very accurate.
Additionally, it has been illustrated that the intracellular drug release from the bioconjugates is required for their antitumor activity. Tumor cell lysosomes, intracellular organelles that have an internal acidic pH (pH<5.0), play a crucial role in the intracellular drug release. Various linkers have been developed in the preparation of drug delivery systems for targeted cancer chemotherapy [1,2]. The most frequently used linker is hydrazone linker, which can be cleaved under acidic environment in lysosomes but keep stable in blood circulation and normal tissues [3,4].
Taken together, we replaced “the prodrug is inactive in blood circulation and normal tissues, but is active in tumor cells” with “the prodrug is inactive in blood circulation and normal tissues, but is active in tumor cell lysosomes”
- Wu, W.; Luo, L.; Wang, Y.; Wu, Q.; Dai, H.B.; Li, J.S.; Durkan, C.; Wang, N.; Wang, G.X. Endogenous pH-responsive nanoparticles with programmable size changes for targeted tumor therapy and imaging applications. Theranostics 2018, 8, 3038-3058.
- Li, Y.; An, L.; Lin, J.; Tian, Q.; Yang, S. Smart nanomedicine agents for cancer, triggered by pH, glutathione, H2O2, or H2S. Int J Nanomedicine 2019, 14, 5729-5749.
- Dai, L.; Zhang, Q.; Shen, X.; Sun, Q.; Mu, C.; Gu, H.; Cai, K. A pH-responsive nanocontainer based on hydrazone-bearing hollow silica nanoparticles for targeted tumor therapy. Mater. Chem. B 2016, 4, 4594-4604.
- Luo, W.; Wen, G.; Yang, L.; Tang, J.; Wang, J.; Wang, J.; Zhang, S.; Zhang, L.; Ma, F.; Xiao, L.; Wang, Y.; Li, Y. Dual-targeted and pH-sensitive doxorubicin Prodrug-Microbubble complex with ultrasound for tumor treatment. Theranostics 2017, 7, 452-465.
Point 6: Line 101: .... presumably because of FA-mediated specific endocytosis. Why endocytosis? and not FA mediated carrier mediated transport? Is FA-AMA-hyd-DOX enclosed in particles (nanoparticles)?
Response 6: Folate targeting was invented soon after Bart Kamen’s group at the University of Texas Southwestern Medical Center reported that folates entered cells via a receptor-mediated endocytic process [1]. The physiological process that mediates folate-targeted drug delivery is illustrated in Fig.2, exogenous folate–drug conjugates bind to externally oriented FRs located on the plasma cell membrane. Then, the cell membrane collapses around the folate receptor and folate drug conjugate to form early endosomes or "vesicles", that is, endocytosis occurs. Then, the lumen of mature endosomes is acidified, the configuration of the receptor is changed, the folate drug conjugate is released, and the breakable bond between the drug and folate is broken and the drug is released [2]. Studies have found that both folate and folate conjugates can enter cells through folate receptor-mediated endocytosis [3-5].
- Kamen, B.A.; Capdevila, A. Receptor-mediated folate accumulation is regulated by the cellular folate content. Proc Natl Acad Sci U S A 1986, 83, 5983-5987.
- Leamon, C.P.; Reddy, J.A. Folate-targeted chemotherapy. Adv Drug Deliv Rev 2004, 56, 1127-1141.
- Geersing, A.; de Vries, R.H.; Jansen, G.; Rots, M.G.; Roelfes, G. Folic acid conjugates of a bleomycin mimic for selective targeting of folate receptor positive cancer cells. Med. Chem. Lett. 2019, 29, 1922-1927.
- Shan, L.; Zhuo, X.; Zhang, F.; Dai, Y.; Zhu, G.; Yung, B.C.; Fan, W.; Zhai, K.; Jacobson, O.; Kiesewetter, D.O.; Ma, Y.; Gao, G.; Chen, X. A paclitaxel prodrug with bifunctional folate and albumin binding moieties for both passive and active targeted cancer therapy. Theranostics 2018, 8, 2018-2030.
- Yang, Z.; Lin, H.; Huang, J.; Li, A.; Sun, C.; Richmond, J.; Gao, J. A gadolinium-complex-based theranostic prodrug for in vivo tumour-targeted magnetic resonance imaging and therapy. Chem Commun (Camb) 2019, 55, 4546-4549.
Point 7: Line 177: ... "Therefore, the MTD of DOX, AMA-hyd-DOX and FA-AMA-hyd-DOX in female athymic nude mice was, for a single injection, approximately 5, 5 and 10 mg / kg , respectively ". For comparability, please also specify doses in µmol/kg
Response 7: Thank you for your advice. The dose unit “5, 5 and 10 mg/kg” have been replaced by “9.2, 9.2 and 18.4 µmol/kg”, which according to the relationship between mass and molar mass. Both the method and result of the original manuscript have been updated.
Reviewer 2 Report
This article entitled “Enhanced Anti-Tumor Effect of Folate-Targeted FA-AMA-hyd-DOX Conjugate in a Xenograft Model of Human Breast Cancer”, describes FA-AMA-hyd-DOX is a promising tumor-targeted compound for breast cancer therapy. This reviewer feels this is an interesting study in the breast cancer research with all the necessary data. Hence this reviewer suggests acceptance of this article to molecules with minor revision.
The minor points are,
- There are some grammatical errors which authors should check carefully and modify the manuscript accordingly. eg. Line 98 “As showed” should be “As shown” etc..
- Page 3, para about the cellular uptake and nuclear localization is not quite clear. As the later part authors have mentioned it is in the cytoplasm. This para should be fixed about the localization which is a very important point in this study.

Author Response
Dear editors and reviewers,
Thank you very much for your letter and advice. We have revised the paper (Enhanced anti-tumor effect of folate-targeted FA -AMA-hyd-DOX conjugate in a xenograft model of human breast cancer, Manuscript ID: molecules-1462525), and would like to re-submit it for your consideration. We have addressed the comments raised by the reviewers, and the amendments are highlighted in red in the revised manuscript. Because of your suggestions, the revised article becomes better and readers can get more valuable information. We hope that the revision is acceptable, and I look forward to hearing from you soon.
With best wishes,
Yours sincerely,
Weiliang Ye
E-mail: yaojixue@fmmu.edu.cn
Comments and Suggestions for Authors,
This article entitled “Enhanced Anti-Tumor Effect of Folate-Targeted FA-AMA-hyd-DOX Conjugate in a Xenograft Model of Human Breast Cancer”, describes FA-AMA-hyd-DOX is a promising tumor-targeted compound for breast cancer therapy. This reviewer feels this is an interesting study in the breast cancer research with all the necessary data. Hence this reviewer suggests acceptance of this article to molecules with minor revision.
The minor points are,
Point 1: There are some grammatical errors which authors should check carefully and modify the manuscript accordingly. eg. Line 98 “As showed” should be “As shown” etc...
Response 1: The whole manuscript was re-checked by an English professor, and all mistakes were corrected in the revised manuscript and marked in red colour.
Point 2: Page 3, para about the cellular uptake and nuclear localization is not quite clear. As the later part authors have mentioned it is in the cytoplasm. This para should be fixed about the localization which is a very important point in this study.
Response 2: Thank you for your advice. The intracellular distribution of FA-AMA-hyd-DOX in MDA-MB-231 cancer cells has been added in the manuscript to illustrate the localization of the DOX.
Figure 1 showed the distribution of DOX red fluorescence in cytoplasm and nucleus after MDA-MB-231 cancer cells were incubated with FA-AMA-hyd-DOX. There was plenty of DOX red fluorescence localized in cytoplasm after MDA-MB-231 cells were incubated for 30 min, while little DOX red fluorescence localized in nucleus. However, a large amount of DOX red fluorescence was localized in nucleus after 2 h incubated, and the DOX red fluorescence in cytoplasm was significantly reduced. These results clearly indicate that the folate receptor-mediated endocytosis process promotes the cellular uptake of FA-AMA-hyd-DOX. Furthermore, when FA-AMA-hyd-DOX was incubated with MDA-MB-231 cells, it was mainly localized in endolysosomes at 30 min. FA-AMA-hyd-DOX showed a distinct nucleus distribution of DOX at 2 h. This is because FA-AMA-hyd-DOX can be dissociated in the endolysosomes, subsequently faster release of DOX and then massively entered into the nucleus to exert cytotoxic effects.